# Profiling serum antibodies with a pan allergen phage library identifies key wheat allergy epitopes

Daniel R. Monaco[1], Brandon M. Sie [2], Thomas R. Nirschl[1], Audrey C. Knight[1], Hugh A. Sampson[3], Anna Nowak-Wegrzyn[3], Robert A. Wood[4], Robert G. Hamilton[5], Pamela A. Frischmeyer-Guerrerio[6,7] & H. Benjamin Larman [1,7]

Allergic reactions occur when IgE molecules become crosslinked by antigens such as food proteins. Here we create the 'AllerScan' programmable phage display system to characterize the binding specificities of anti-allergen IgG and IgE antibodies in serum against thousands of allergenic proteins from hundreds of organisms at peptide resolution. Using AllerScan, we identify robust anti-wheat IgE reactivities in wheat allergic individuals but not in wheat-sensitized individuals. Meanwhile, a key wheat epitope in alpha purothionin elicits dominant IgE responses among allergic patients, and frequent IgG responses among sensitized and non-allergic patients. A double-blind, placebo-controlled trial shows that alpha purothionin reactivity, among others, is strongly modulated by oral immunotherapy in tolerized individuals. AllerScan may thus serve as a high-throughput platform for unbiased analysis of anti-allergen antibody specificities.

[1] Institute for Cell Engineering, Division of Immunology, Department of Pathology, Johns Hopkins School of Medicine, Baltimore, MD, USA. [2] Bioinformatics and Integrative Genomics PhD Program, Harvard Medical School, Boston, MA, USA. [3] Icahn School of Medicine at Mount Sinai, New York, NY, USA. [4] Johns Hopkins University School of Medicine, Johns Hopkins Hospital, Baltimore, MD, USA. [5] Division of Allergy and Clinical Immunology, Department of Medicine, and Department of Pathology, Johns Hopkins University School of Medicine, Baltimore, MD 21205, USA. [6] The Laboratory of Allergic Diseases, National Institutes of Allergy and Infectious Diseases, Bethesda, MD, USA. [7] These authors jointly supervised this work: Pamela A. Frischmeyer-Guerrerio, H. Benjamin Larman. ✉email: pamela.guerrerio@nih.gov; hlarman1@jhmi.edu

In recent decades, food allergy has emerged as a major public health issue, affecting up to 10% of the population in westernized countries[1]. In patients with IgE-mediated food allergy, exposure to allergenic food results in cross-linking of preformed food-specific IgE (fs-IgE) bound to the high-affinity IgE receptor FcεRI on the surface of mast cells and basophils, causing potentially life-threatening allergic reactions. The high prevalence of food allergy has led to an ever-increasing need for food allergy testing in clinical practice. While oral food challenge is the gold standard for diagnosing food allergy, this procedure is time-consuming, requires highly trained personnel, and can cause an acute allergic reaction. Therefore, the diagnosis of food allergy is often based on a combination of patient clinical history and the results of fs-IgE testing and skin prick testing. These tests, however, often detect sensitization to foods that are not associated with symptoms upon ingestion, which can lead to unnecessary food avoidance. The shortcomings of fs-IgE testing are exemplified in the diagnosis of IgE-mediated wheat allergy, where a meta-analysis found that wheat-specific IgE levels have a specificity of only 43% in predicting wheat allergy[2]. Strong cross-reactivity between grass pollen and wheat is likely a contributing factor to the high rate of false-positive tests[3].

While component testing has emerged as a useful adjunct for diagnosing allergy for select foods (peanut, hazelnut), this approach is limited to testing single allergens. Comprehensive characterization of allergies via component testing requires relatively large sample volumes and significant expense. Other strategies to refine the diagnosis of food allergy have relied on epitope mapping, which evaluates IgE binding to a library of contiguous short peptides that compose allergenic proteins. Several methodologies have been developed, including SPOT membranes, microarray based immunoassays, and most recently Bead-Based Epitope Assays[4,5]. These approaches have revealed that certain immunodominant peptides, as well as overall greater diversity of IgE epitopes recognized, are associated with more severe reactions and a greater likelihood of having persistent allergy in patients with milk and egg allergy[4].

However, these approaches are largely only capable of assessing antibody reactivities to a select number of known allergenic components (typically one to a few dozen) and are frequently difficult to interpret. Importantly, the identification of novel allergenic epitopes is both costly and time-intensive; inexpensive high-throughput approaches that efficiently identify novel epitopes would therefore have great utility to inform clinical component and allergenic epitope test development.

In this work, we present a programmable phage display based method to comprehensively analyze anti-allergen IgE and IgG antibodies to 1847 allergenic proteins, tiled with overlapping 56 amino acid peptides, in a single multiplex reaction. Similar to other Phage ImmunoPrecipitation Sequencing (PhIP-Seq) libraries[6–8] we use oligonucleotide library synthesis to encode a database of allergenic peptide sequences for display on T7 bacteriophages (the "AllerScan" library), which can be analyzed using high-throughput DNA sequencing. While lacking in highly conformational, discontinuous, and post-translationally modified epitopes, PhIP-Seq enables high-resolution analysis of longer, higher quality peptides than is otherwise possible with synthetic peptide microarrays, and at a dramatically reduced per-sample cost.

Using the AllerScan phage display library, we identify thousands of IgE and IgG antibody reactivities against hundreds of distinct organisms with peptide level resolution. We identify robust differences in the anti-wheat IgE reactivities of wheat allergic individuals compared to both non-allergic and wheat-sensitized individuals. Finally, we identify a wheat allergen epitope in alpha purothionin that may prove to be particularly useful in distinguishing wheat allergy from wheat sensitization.

AllerScan is a massively multiplexed serological assay with utility for correlating clinical characteristics with anti-allergen IgE and IgG reactivity at cohort scale.

## Results

**AllerScan library**. In order to construct a comprehensive library of allergenic proteins, the curated Allergome database[9] was downloaded from UniProt (accessed 6 August 2017) and used as input to the PhIP-Seq pepsyn library design pipeline[7]. The 1847 proteins of the Allergome database were represented as a set of 19,332 56 amino acid peptide tiles with 28 amino acid overlaps, which were encoded by a library of synthetic 200-mer oligonucleotides (Fig. 1a). This library was amplified and cloned into the T7 phage display system[6] for automated serological profiling[7]. We refer to this library as the T7 AllerScan library.

We have previously utilized protein A and protein G-coated magnetic beads to immunoprecipitate predominantly IgG-bound phage[7]. In contrast to the highly abundant and relatively consistent levels of IgG in blood, IgE tends to be logs lower in abundance and highly variable between individuals[10]. While it has been reported that protein A does have a minor affinity to IgE[11], protein A's binding affinity to IgG is significantly more robust. This difference in binding affinity coupled with the difference in abundance between serum IgG vs IgE levels suggest potential binding between protein A and IgE would be negligible. To enable specific IgE immunoprecipitation of the AllerScan library, we covalently conjugated biotin to the therapeutic monoclonal anti-IgE antibody omalizumab[12]; streptavidin coupled magnetic beads could then be irreversibly coated with this IgE capture antibody (Fig. 1a).

To assess the quality of the AllerScan library, we PCR amplified and sequenced the post-expansion phage library at 66-fold coverage. We detected sequences from 95.8% of the 19,331 unique clones in the AllerScan library, and found they were drawn from a relatively uniform distribution (Fig. 1b). Next, we serially diluted serum from two individuals, one with a known IgE-mediated wheat allergy and one with a known IgE-mediated peanut allergy. In both cases, AllerScan revealed concentration-dependent enrichments of the expected allergenic peptides in the immunoprecipitated fraction (Fig. 1c, d). Based on these data, 100 ng of total, volume normalized IgE input was chosen for subsequent analyses. Next, to establish the reproducibility and specificity of the IgE immunocapture procedure, we performed two IgE and two protein A/G (hereafter referred to as IgG) immunocaptures using serum from a wheat allergic individual. We noted a high concordance when comparing replicas from the same immunocapture technique (IgE: $R^2 = 0.996$, Fig. 1e), and a high level of discordance when comparing IgE to IgG (IgE vs IgG: $R^2 = 0.0157$, (Fig. 1f). This discordance, particularly for the strongest reactivities, highlights the isotype specificity of omalizumab-based IgE immunocapture.

**IgE profiling with AllerScan**. We next assembled a cross-sectional cohort of individuals with clinically characterized IgE-mediated food allergies including wheat and/or peanut, as well as age-matched healthy controls. Additional clinical metadata on these individuals are displayed in Supplementary Table 1. Figure 2a, b displays the results of both IgE (Fig. 2a) and IgG (Fig. 2b) immunocaptures. In these heatmaps, each column corresponds to a peptide that reacted with at least three sera. The order of the columns corresponds to the taxonomic identity of the organism from which each peptide is derived. Groups of related organisms, rather than individual organisms, are indicated for clarity. Group membership is displayed in Supplemental Table 2. Rows correspond to individual samples and were grouped by wheat allergy.

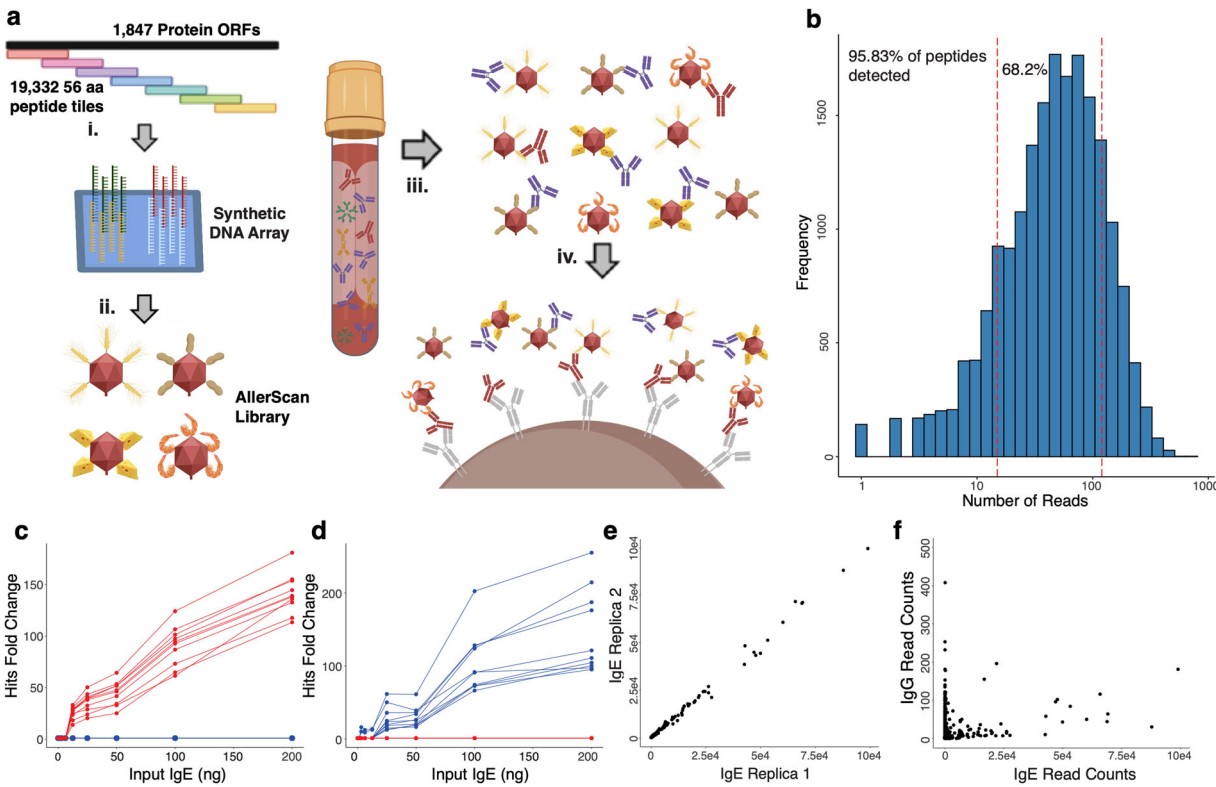

**Fig. 1 IgE profiling with the T7 phage-displayed AllerScan library. a** (i) The AllerScan library is composed of 19,331 56 aa peptide tiles that overlap by 28 aa. (ii) The peptide-encoding DNA sequences are synthesized as 200-mer oligonucleotides and cloned into a T7 phage display vector. (iii) The AllerScan phage library is incubated with serum. (iv) IgE and bound phage are immunocaptured onto omalizumab-coated magnetic beads and sequenced. **b** Representation of the AllerScan phage library: 95.84% of all library members are detected; 68.2% of the library members are within one log of abundance. **c** Dilution series showing top 10 wheat peptides (red) and peanut peptides (blue) using serum from a wheat allergic, peanut-tolerant individual. **d** Dilution series showing top 10 wheat peptides (red) and peanut peptides (blue) using serum from a peanut allergic, wheat-tolerant individual. **e** Reproducibility of IgE PhIP-Seq assay. **f** Discordance of one individual's IgE and IgG reactivity against the AllerScan library. Source data are provided as a Source Data file.

Each group was organized by hierarchical clustering of IgE profiles. The order of the rows and columns is maintained for the IgG heatmap (Fig. 2b). Reactivity to wheat peptides were observed to correlate with the expected allergic group, while many additional allergic reactivities were also observed. Figure 2c highlights the change in the library's peanut and wheat representation in the immunoprecipitated fractions, compared to the input (starting) AllerScan library. As expected for this cohort, both wheat and peanut showed large increases in representation post-immunoprecipitation.

**Wheat IgE and IgG specificity analysis**. Since few studies have investigated anti-wheat reactivity at the epitope level, we sought to broadly characterize the fine specificities of anti-wheat antibodies using the AllerScan system. To this end, we compared both IgE and IgG immunocaptures across serum samples from three groups: individuals with IgE-mediated wheat allergy (positive wheat-specific IgE testing by immunoCAP and convincing history of an acute IgE-mediated reaction during a recent home exposure or clinic wheat food challenge; $N = 32$), individuals with wheat "sensitivity"[13] (sensitized; detectable wheat-specific IgE testing but tolerating wheat in their diet; $N = 27$), and non-allergics (undetectable wheat IgE and tolerating wheat in their diet; $N = 27$). An expansive set of 577 wheat peptides was found to be commonly IgE reactive in the wheat allergic group out of a total 1525 wheat-associated peptides in the AllerScan library. Minimal reactivity was detected to the phage-displayed wheat peptides in sera from either sensitized or non-allergic individuals (Fig. 3a).

Among wheat allergic IgE responses, we noted both extensive immunodominance and distinct patterns of reactivity. Upon examination of IgG reactivity to wheat peptides among the same three groups, we noted dramatic elevation of reactivity among allergic individuals—largely to the same IgE immunodominant epitopes, along with detectable but relatively less reactivity among the sensitized individuals, and minimal reactivity among the non-allergic individuals (Fig. 3b). Several previously identified wheat epitopes are composed of repetitive and redundant peptide sequences[14–16]. We therefore utilized a network graph-based approach to determine, for each individual immunocapture, the "maximal independent vertex set" of reactive peptides that do not share any sequence homology. We have previously utilized this metric as a surrogate for the "breadth" of a polyclonal response[17,18]. Wheat allergic individuals exhibited significantly greater breadths of both IgE and IgG reactivity (Fig. 3c, d) versus non-allergic individuals ($p = 2.5 \times 10^{-11}$ for IgE and $p = 1.9 \times 10^{-4}$ for IgG). Only in the IgG immunocaptures were the breadths of responses to wheat peptides significantly higher in the sensitized versus the non-allergic individuals ($p = 0.0034$). Taken together, these findings suggest that the breadth of the anti-wheat peptide IgE repertoire may have utility in distinguishing between wheat allergy and sensitization.

**Anti α-purothionin reactivity**. Upon examination of the wheat peptide reactivity matrix, we noted that, whereas allergic individuals displayed higher IgG reactivity for nearly every dominant epitope, a seemingly distinct set of peptides (Fig. 3b, arrows) were frequently recognized by both non-allergic and sensitized individuals' IgG, but infrequently by allergic individuals' IgG. These

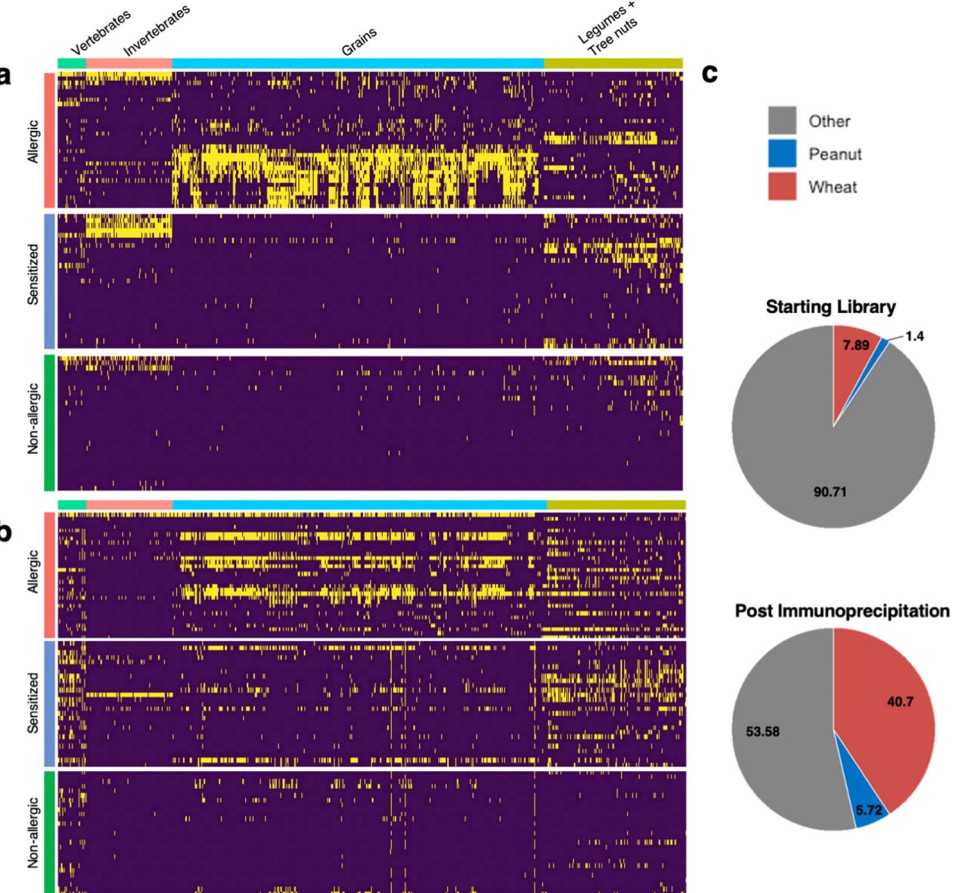

**Fig. 2 AllerScan profiles.** IgE (**a**) and IgG (**b**) reactivity profiles for a cross-sectional set of test samples ($n = 86$). Columns correspond to peptides recognized by at least three samples; reactivities to organisms with less than five recognized peptides were excluded. Columns are arranged taxonomically via phylogenetic clustering. Each row corresponds to a unique sample. IgE rows were grouped by wheat allergy status and each subgroup was hierarchically clustered via a binary distance metric; the IgE row and column order was maintained for the IgG reactivity profiles. **c** Pie charts corresponding to the library's peanut and wheat representation in the starting library (top) and in the immunoprecipitated fractions (bottom). Source data are provided as a Source Data file.

peptides were all derived from two regions of the alpha pur-othionin protein, which is also known as Tri a 37, and present as three distinct isoforms in the AllerScan library[19]. The N-terminal region contains a dominant epitope (Fig. 4a, b), while the C-terminal region contains a sub-dominant epitope rarely recognized on its own. IgE reactivity to the dominant alpha purothionin motif was frequent among allergic individuals, but almost never detected among the non-allergic or sensitized individuals (Fig. 4c). These data suggest that the relative amount of IgE versus IgG anti-alpha purothionin reactivity may also have utility in distinguishing allergic from sensitized individuals (Fig. 4d).

**Wheat allergy subgroup analysis**. We next asked whether there were distinguishable patient and peptide subgroups within the wheat allergic cohort based upon their patterns of anti-wheat IgE reactivity profiles. To this end, the IgE wheat reactivity data were subsetted to include only peptides recognized by at least four wheat allergic individuals and subjected to hierarchical clustering, which revealed six distinct peptide clusters and three patient populations (Fig. 5a). Two of these peptide clusters were composed entirely of high molecular weight (HMW) glutenin pep-tides, one group was composed almost entirely of low molecular weight (LMW) glutenin peptides, two other groups were composed primarily of alpha, gamma, and omega gliadin peptides and

the final group was composed primarily of alpha purothionin peptides. At the population level, patient cluster 2 was the only cluster to recognize HMW glutenin, whereas patients in cluster 2 and cluster 3 both demonstrated reactivities to LMW glutenin. Groups two and three had significantly higher anti-wheat IgE breadth than group 1 (Fig. 5b). We additionally investigated potential differences in IgE and IgG reactivity to specific peptides (Fig. 5c) and found overall discordance in peptide reactivity at the antibody isotype level, with more than 80% of all peptides dis-playing exclusively IgE or IgG reactivity.

We explored whether the observed clustering of peptide reactivities might be explained at least in part by peptide sequence homology. A peptide sequence alignment-based net-work graph was therefore constructed, with the peptide nodes colored according to the protein from which they were derived (Fig. 5d). This graph separated into three disconnected subgraphs: one cluster of HWM glutenin peptides, one cluster composed of gliadins and LMW glutenins, and one cluster composed exclusively of alpha purothionin. There were also 22 singleton peptides lacking any alignment to other enriched peptides (not shown). We noted dense families of homologous peptides represented among LMW and HMW glutenin peptides, which reflects the repetitive sequences found among the glutenins[15]. Indeed, all peptides in the HMW glutenin subgraph shared one tightly conserved epitope (Fig. 5d, top left).

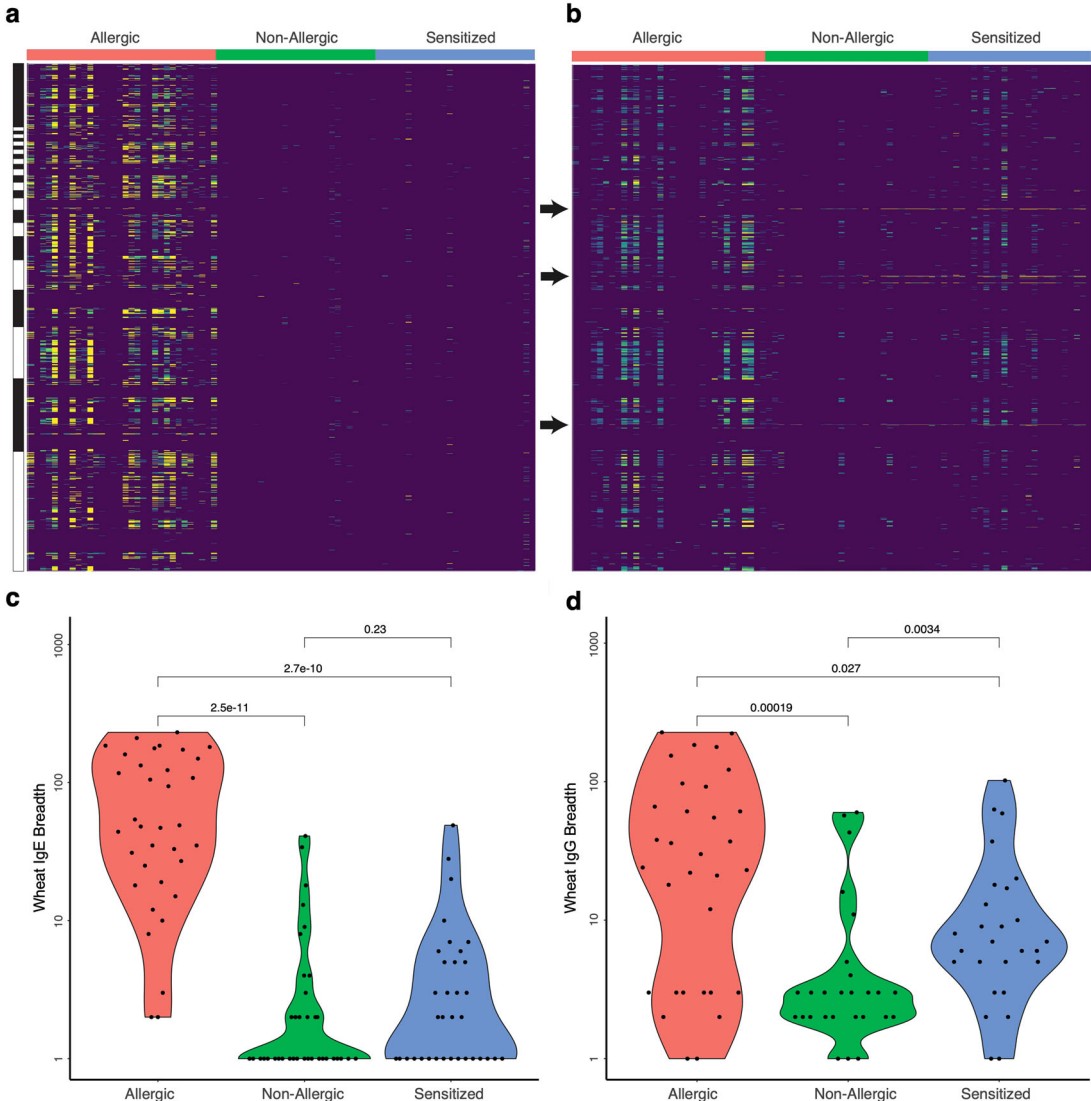

**Fig. 3 IgE and IgG reactivity of wheat peptides.** Serum donors (columns in **a**, **b**) had been clinically defined as wheat allergic, wheat non-allergic, and wheat sensitized. Rows correspond to wheat peptides tiled from each protein's N- to C-terminus, arranged by increasing size (top to bottom). Data in **a** are from IgE profiling; data in **b** are from IgG profiling. **c** Comparison of the IgE anti-wheat antibody breadths between patient groups. **d** Comparison of the IgG anti-wheat antibody breadths between patient. For **b**–**d**—wheat allergic (n = 32), wheat non-allergic (n = 27), and wheat sensitized (n = 27). Significance values in both **c** and **d** were defined via two-sided Wilcox ranked sum test (with no multiple hypothesis test corrections). Source data are provided as a Source Data file.

**Antibody responses to wheat oral immunotherapy**. We next used AllerScan to characterize changes in wheat IgE and IgG reactivity in response to wheat oral immunotherapy (WOIT). Twenty-three participants in the previously described food allergy cohort were also enrolled in a randomized, double-blind, placebo-controlled clinical trial of WOIT[20]. Participants in this trial received either vital wheat gluten OIT or placebo, with biweekly escalations for up to 44 weeks until reaching a daily maintenance dose of 1445 mg of wheat protein or placebo. Serum from participants was assessed at three timepoints: at trial start, after 1 year of placebo treatment (for participants originally enrolled as placebos), and after 1 year of active wheat OIT. In these AllerScan analyses, data are plotted in a pairwise fashion; unchanged reactivities fall along the $y = x$ diagonal, whereas reactivities that increase or decrease over time appear in the upper left or lower right quadrant, respectively. Most individuals in the placebo arm of the trial exhibited minimal changes to their anti-allergen IgE or IgG profiles (see Fig. 6a for representative example). In stark

contrast, however, we observed dramatic wheat-specific alterations in IgE and IgG reactivity during WOIT therapy in many individuals (see Fig. 6b for representative example); as expected, reactivity to non-wheat allergens remained stable over this time period. A trend of decreasing IgE and increasing IgG reactivity to wheat peptides after wheat treatment was observed across this WOIT cohort (see Fig. 6c for aggregated data). Next, we compared anti-wheat IgE and IgG reactivities separately, from all patients, before and after active treatment to discern whether any new reactivities were developing in response to treatment (Fig. S1). Overall, we found that most IgE reactivities were initially present at baseline, although there was a small population of IgE reactivities that were only detected post immunotherapy. We found the opposite trend with IgG reactivities, with most reactivities not appearing until after immunotherapy.

During the placebo phase of the trial, neither anti-wheat breadth nor specific reactivity to alpha purothionin changed significantly, either in IgE or IgG (Fig. 6d). In contrast, the

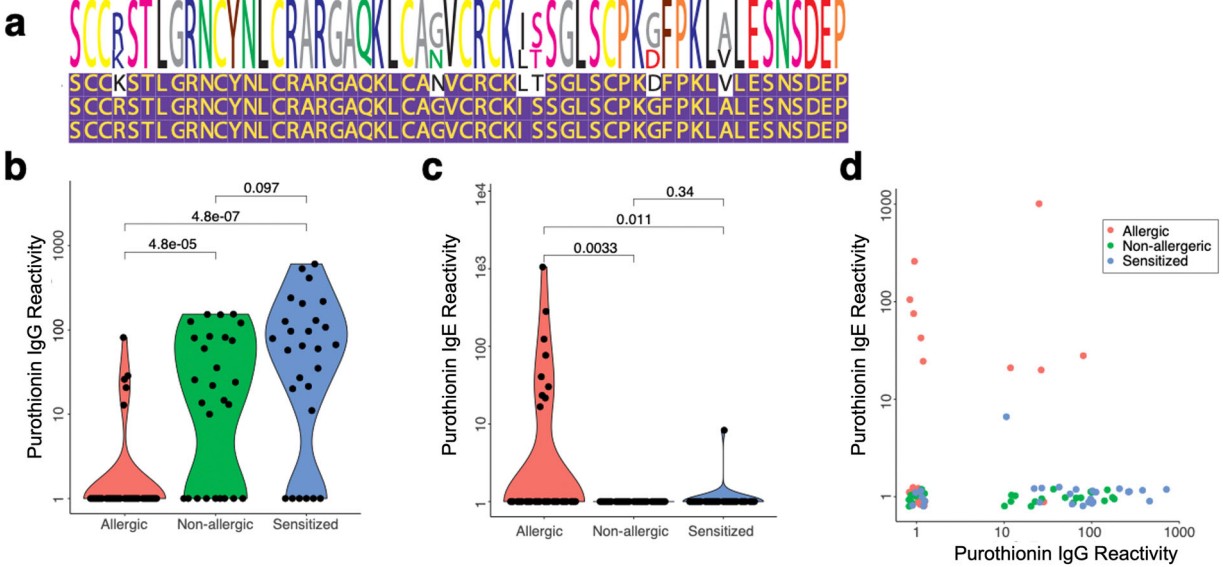

**Fig. 4 Identification of a discriminatory IgG-reactive wheat epitope. a** Multiple sequence alignment of three alpha purothionin peptides, which are **b** preferentially IgG reactive in the wheat non-allergic and sensitized populations, versus wheat allergic individuals. **c** IgE reactivities to the alpha purothionin epitope among the same three patient populations. **d** Comparison of the IgE versus the IgG reactivities to the alpha purothionin epitope. For **b–d**—wheat allergic (n = 32), wheat non-allergic (n = 27), and wheat sensitized (n = 27). Source data are provided as a Source Data file.

breadth of the anti-wheat IgE repertoire decreased significantly in response to WOIT ($p = 0.01$, Fig. 6e), while the specific anti-alpha purothionin IgG response increased dramatically ($p = 0.0006$, Fig. 6e). There was no association between anti-alpha purothionin IgE and WOIT treatment outcome. Taken together, our data point to alpha purothionin IgG reactivity as an important biomarker of both wheat insensitivity and desensitization.

## Discussion

We have created the AllerScan phage display library, which is composed of all protein sequences present in the Allergome database and used it to characterize both IgE and IgG antibody reactivities in a cohort of food allergic individuals. Similar to previous PhIP-Seq libraries, AllerScan employs high-throughput oligonucleotide synthesis to efficiently encode high quality, overlapping 56 amino acid peptide sequences. Currently, polypeptides of this length cannot be reliably synthesized chemically. Here, we used IgE-specific immunoprecipitation of phage-displayed peptides, followed by high-throughput DNA sequencing to identify reactive peptides. Incorporation of liquid-handling automation, as we have done in this study, increases sample throughput and enhances assay reproducibility. High levels of sample multiplexing can be achieved by incorporating DNA barcodes into the PCR amplicon, prior to pooling and sequencing; at a sequencing depth of ~10-fold, ~500 AllerScan assays can be analyzed on a single run of an Illumina NextSeq 500. Batched analyses can therefore bring down the assay cost to just a few dollars per sample.

PhIP-Seq with the AllerScan library provides quantitative antibody reactivity data at epitope level for about 2000 proteins derived from hundreds of organisms. This unbiased approach evaluates antibody reactivities to major and minor allergens, including both well-defined and poorly studied epitopes. The library can be readily updated by simply adding new allergens as they are identified. Future studies using additional libraries, such as the human or VirScan phage libraries[6,8], will enable a broader analysis of IgE reactivity.

Limitations of all programmable phage-based assays include the lack of post-translational modifications and discontinuous epitopes. While these two types of epitopes are likely critical for certain allergens[21], protein denaturation during digestion is thought to reduce their functional significance for food allergies[22]. It is therefore possible that the AllerScan library may exhibit reduced sensitivity for non-food allergic antibodies. This topic will be addressed in future work.

In this study we utilized a constant input of total IgE, versus sIgE, to explore the unbiased utility of AllerScan for identifying any potential allergic antibody reactivity. Normalizing by sIgE input may be useful in future studies focused on specific allergens. We wondered, however, whether the amount of wheat sIgE confounded our analysis of allergic versus sensitized reactivities. Supplementary Fig. 2 provides AllerScan outcome data as a function of wheat sIgE, separately for the allergic and the sensitized individuals. These data indicate that for allergic individuals, greater sIgE in the reaction does associate with greater AllerScan reactivity. For sensitized individuals, however, greater sIgE titer does not correlate with greater AllerScan reactivity.

Here we analyzed over one million antibody–allergen peptide interactions in a comprehensive study of pan-allergen serology from a cross-sectional cohort of patients with peanut and/or wheat allergies, as well as non-allergic controls. Peanut and wheat allergens were the most widely recognized in this cohort, but we also detected reactivities against many other allergen sources including tree nuts, invertebrate tropomyosin, and milk proteins. This observation is not unexpected since many of the participants in our study had additional food allergies, which is often the case in patients with food allergy in the general population[23]. The diagnosis of IgE-mediated wheat allergy is particularly challenging as wheat-specific IgE levels do not reliably predict allergy. Although a number of major wheat allergens have been identified in patients with IgE-mediated wheat allergy, IgE component testing for these proteins has not reached the sensitivity and specificity necessary to incorporate them into clinical practice. We sought to characterize anti-wheat antibody responses due to the current gap in knowledge about their relevant fine specificities. Our results suggest that peptide-level IgE reactivity may enhance discrimination between wheat allergy and sensitization. We also found that sensitized (but not allergic) individuals

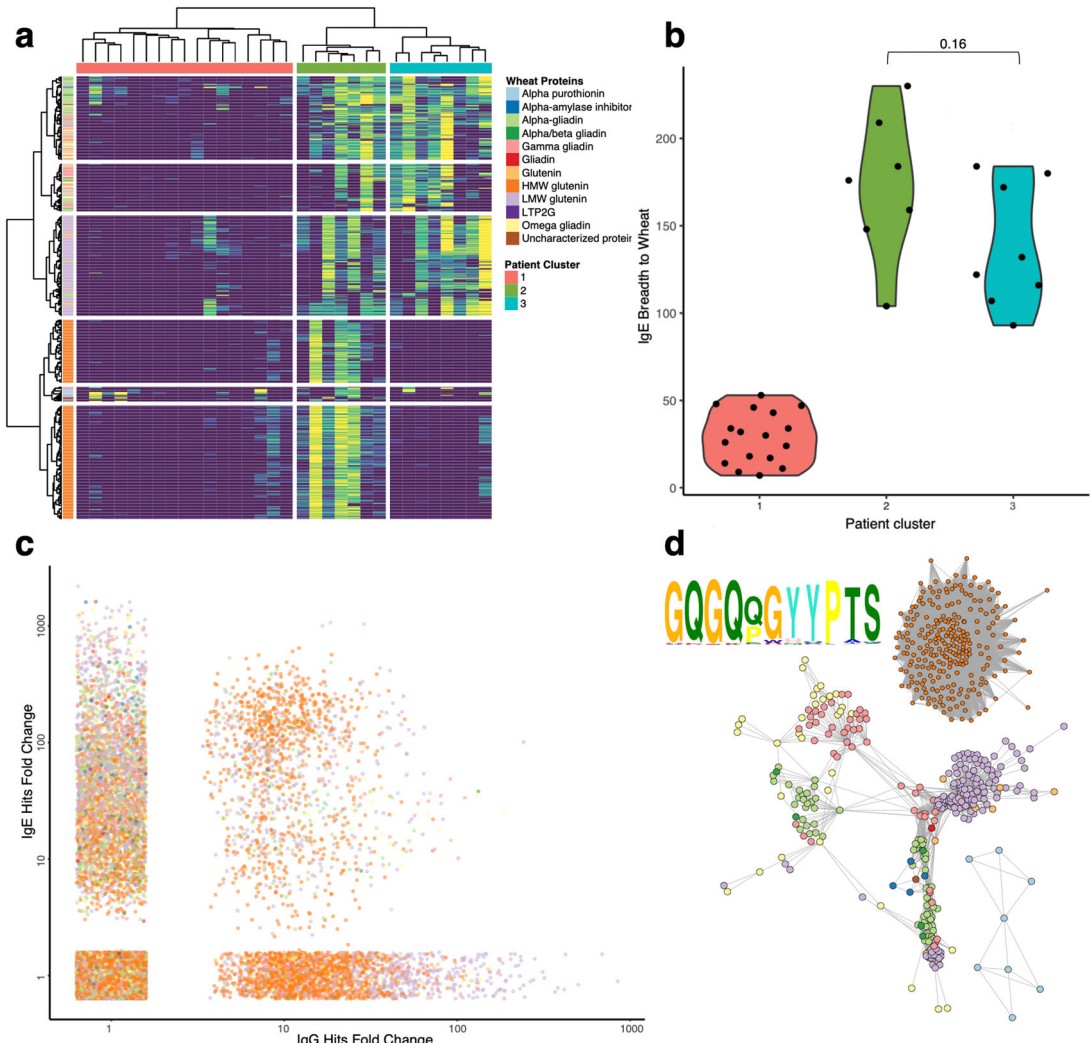

**Fig. 5 Identification of wheat allergic antibody subgroups. a** Hierarchically clustered heatmap of IgE reactivity to wheat peptides among allergic individuals ($n = 32$). Rows are wheat peptides recognized by at least four wheat allergic sera. **b** Overall wheat IgE antibody breadth between clusters (cluster 1—$n = 17$, cluster 2—$n = 7$, cluster 3—$n = 8$). All comparisons between populations were performed via two-sided Wilcoxon signed-rank test (with no multiple hypothesis test corrections). **c** Scatterplot comparing IgE vs IgG reactivity to all peptides from **a**. Each point compares a given antibody reactivity for a given allergic individual. **d** Network graph of all peptides in **a**. Nodes are peptides; nodes are linked if they share sequence similarity. A multiple sequence alignment logo generated from the HMW glutenin cluster is shown in the top left. The color scheme for **c** and **d** is conserved from **a**. Source data are provided as a Source Data file.

harbored elevated anti-wheat IgG responses compared to their non-allergic counterparts. These data are consistent with the possibility that sensitized individuals also harbor IgE responses to wheat, but that compensatory IgG can effectively block pathogenic IgE binding to wheat epitopes[24].

We observed consistent patterns of wheat reactivity among the allergic cohort; some peptides were reactive in over 80% of individuals with wheat allergy, whereas most peptides did not react with sera from any individual included in this study. Dominant wheat reactivities tended to occur in highly repetitive regions of allergenic proteins; this phenomenon has also been described for other allergens[25]. Additionally, we compared isotype differences in wheat peptide reactivity and found notable discordance in IgE and IgG reactivity for most peptides, as has been observed in other epitope studies; only 16% of reactive peptides were bound by both IgE and IgG isotypes. It is possible that this discordance may be partially explained by competition between the two antibody classes. Future experiments in which purified IgE and IgG are used for AllerScan analysis will test this hypothesis. Future studies using AllerScan may also provide insight into questions about B cell development, class-switching, and affinity maturation, particularly when used in high-resolution longitudinal studies or in animal models of allergic disease.

In wheat allergic individuals, the most reactive wheat epitopes were derived from alpha, beta, gamma, and omega gliadin and both HMW and LMW glutenin . These proteins all harbor highly repetitive domains[21]. We used sequence alignment to characterize homology among all dominant epitopes, which revealed modest homology between all gliadin peptides. Reactive omega and gamma gliadin epitopes were particularly homologous, which has been previously reported[26]. IgE reactive LMW glutenin peptides also exhibited modest sequence homology with the gliadins, but did not share homology with HMW glutenin peptides. The reactive peptides derived from HMW glutenin were characterized by a small number of highly repetitive motifs. All reactive HMW glutenin peptides possessed at least one IgE-binding epitope with strong consensus to previously reported HMW glutenin epitopes[21].

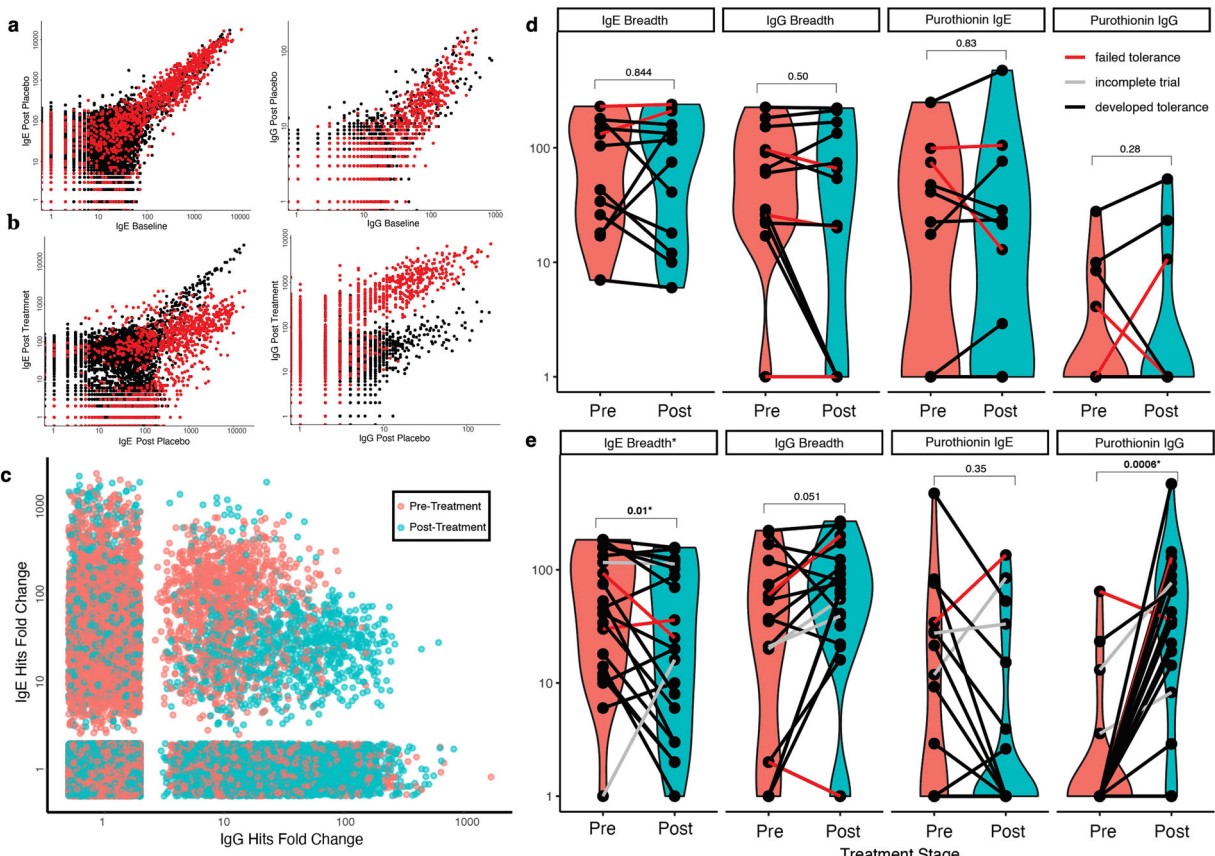

**Fig. 6 AllerScan analysis of wheat oral immunotherapy trial. a** Pairwise plots of longitudinal IgE (left) and IgG (right) AllerScan read count data from one patient after 1 year of receiving placebo. Wheat peptides are red, all other peptides in the AllerScan library are black. **b** Pairwise plots of IgE (left) and IgG (right) AllerScan read count data from same patient as in **a**, comparing serum prior to WOIT treatment against serum after 1 year of WOIT treatment. **c** Scatterplot comparing IgE vs IgG reactivity to wheat peptides from Fig. 4, before and after WOIT treatment. **d** Violin plots evaluating overall wheat antibody breadth and alpha purothionin reactivity before and after receiving placebo treatment ($n = 13$). No significant changes were observed for any metric after placebo (two-sided Wilcoxon signed-rank test). **e** Violin plots evaluating overall wheat antibody breadth and alpha purothionin reactivity before and after receiving WOIT treatment ($n = 21$). There was a significant decrease in IgE wheat breadth and a significant increase in IgG alpha purothionin reactivity after treatment ($p < 0.01$ for both comparisons, two-sided Wilcoxon signed-rank test). Source data are provided as a Source Data file.

Alpha purothionin is a plant defense protein abundantly expressed in wheat seeds[19]. IgE that recognizes this protein has been associated with a fourfold increased risk of experiencing severe allergic reactions to wheat[27]. Interestingly, while we did not detect an association between allergy severity and alpha purothionin reactivity in our study, we did identify anti-alpha purothionin IgG reactivity as the most prevalent (72%) reactivity in non-allergic and sensitized individuals, whereas IgG reactivity was low among individuals with wheat allergy. Conversely, IgE reactivity to alpha purothionin was found in 28% of wheat allergic individuals and only 2% of non-allergic/sensitized individuals. IgG and IgE reactivity to alpha purothionin may therefore be quite useful in distinguishing between allergy and sensitization to wheat.

The quantitative nature of AllerScan data enabled a longitudinal analysis of response to WOIT in a placebo-controlled, double-blind study involving 23 wheat allergic individuals. In response to WOIT, we noted a dramatic shift from IgE to IgG reactivity with wheat peptides, whereas reactivity to other allergens was not affected. As expected, individuals experienced no change in anti-wheat reactivity while in the placebo arm of the trial. Nearly every participant receiving treatment exhibited an overall reduction in anti-wheat IgE repertoire breadth and a concurrent increase in IgG repertoire breadth, an observation that has been reported in other food oral immunotherapy trials[28].

Interestingly, most (82.6%) allergic patients receiving WOIT experienced an increase in IgG reactivity to alpha purothionin. We additionally compared changes in IgE and IgG wheat reactivity with clinical responses and found no statistically significant associations after multiple hypothesis test correction; a future, more sufficiently powered, study may identity significant correlations.

In conclusion, AllerScan is a approach for epitope-level characterization of allergen-associated IgE and IgG responses at cohort scale. We have demonstrated its utility for broad pan-allergen analysis as well as for characterizing the fine specificities of anti-wheat antibodies. Our initial research has confirmed numerous previously known properties of wheat allergy and has revealed reactivity towards an epitope that discriminates between wheat allergy and sensitization. AllerScan is therefore a valuable tool for allergy research and may ultimately provide a powerful adjunct to current methods for diagnosing food allergy and monitoring response to immunotherapy.

## Methods

**Study participants**. Sera for AllerScan were obtained from 58 patients with IgE-mediated food allergy along with 25 age-matched healthy controls who were enrolled on a Natural History of Food Allergy protocol at the National Institutes of Health (NIH). Clinical metadata on the participants are shown in Supplementary Table 1. Participants were defined to have wheat allergy if they had a convincing history of a type I hypersensitivity reaction to wheat within the last 2 years

immediately after ingesting wheat along with positive wheat-specific IgE testing, with the exception of three participants who were classified as wheat allergic even though their most recent reaction was 5 and 6 years ago, and one who had been avoiding wheat due to positive testing. We used the ImmunoCAP assay with a lower limit of detection of 0.35 kUa/L to detect sensitization/allergy. Wheat-sensitized individuals were tolerating wheat in their diet with no overt symptoms, and all non-allergic controls were following an unrestricted diet. Peanut allergy was similarly defined by a positive immediate reaction upon peanut ingestion along with positive peanut-specific IgE testing, although 13 individuals were avoiding peanut due to positive testing alone. Peanut and wheat-specific IgE levels and total IgE were determined by ImmunoCAP (Phadia). This study was approved by the institutional review board of the National Institutes of Health and participants underwent informed consent and assent.

Sera were also evaluated from 23 wheat allergic individuals who participated in a randomized, double-blind, placebo-controlled WOIT trial. Details of the clinical study are described in ref. [20]. Briefly, participants with wheat allergy confirmed by a positive double-blind, placebo-controlled oral food challenge (DBPCFC) to wheat at baseline were randomized 1:1 to OIT with vital wheat gluten or placebo. Participants underwent dose escalation every 2 weeks until they reached a daily maintenance dose of 1445 mg of wheat or placebo protein. After approximately 1 year of treatment (minimum of 8 weeks of maintenance dosing), participants underwent a DBPCFC to wheat (cumulative dose of 7443 mg wheat protein) to evaluate for desensitization and were subsequently unblinded. Participants in the active arm continued on active wheat OIT, while the placebo arm crossed over to active treatment (maximum maintenance dose of 2748 mg wheat protein) for approximately 1 year. Individuals evaluated by AllerScan were enrolled in the WOIT trial at Johns Hopkins University and the Icahn School of Medicine at Mount Sinai in New York. The WOIT trial was approved by both institutional review boards and informed consent was obtained from all participants or their parents. All primary outcomes of the trial were published at clinicaltrials.gov (NCT01980992).

**Construction of the AllerScan phage display library**. The AllerScan library was constructed following the previously detailed PhIP-Seq protocol[7] Briefly, we downloaded all protein sequences from Uniprot database included in the Allergome and collapsed on 90% sequence identity [https://www.uniprot.org/uniref/?query=uniprot:(allergome)+identity:0.9]. The UniProt clustering algorithm returned representative sequences for each protein cluster. We designed peptide sequences 56 amino acids in length that overlap by 28, to tile through all representative allergen proteins. Next, we reverse-translated these sequences into DNA nucleotide sequences optimized for *Escherichia coli* expression and added adapter sequences to both the 5′ and 3′ ends. These 200 nucleotide sequences were synthesized by Twist Bioscience. This oligonucleotide library was PCR amplified and then restriction cloned into the T7FNS2 vector backbone. The resulting library was packaged via the T7 Select Packaging Kit (EMD Millipore) and then expanded for use as a PhIP-Seq library.

**Phage ImmunoPrecipitation Sequencing**. IgG screening of the AllerScan library was performed as described previously[6–8]. Briefly, an IgG-specific enzyme-linked immunosorbent assay was used to quantify serum concentrations (Southern Biotech). Two micrograms of IgG was added to 1 mL of AllerScan library at $1.9 \times 10^9$ pfu for each reaction. Serum and phage library were rotated overnight at 4 °C, after which 20 µL of protein A and 20 µL of protein G-coated magnetic beads (Invitrogen,10002D and 10004D) were added to each reaction, which was rotated an additional 4 h at 4 °C. Beads were subsequently washed three times in 0.1% NP-40 and then resuspended in a Herculase II Polymerase (Agilent cat # 600679) PCR master mix. These PCR reactions underwent 20 cycles of PCR. Two microliters of each reaction was added to a new PCR reaction containing sample-specific barcoding primers. These reactions underwent an additional 20 cycles of PCR. The final amplified product was pooled and sequenced using an Illumina NovaSeq 6000 or NextSeq 500.

For IgE Phip-Seq, biotinylated omalizumab (Labome—KBI1021) was conjugated to Streptavidin-bound M280 magnetic beads (ThermoFisher—11205D) at 4× binding capacity. Excess unbound omalizumab was washed away with 0.01% PBST. One hundred nanograms of IgE was added to 1 mL of AllerScan library at $1.9 \times 10^9$ pfu for each reaction and 10 µL of omalizumab-coated M280s was used for each immunoprecipitation. All other steps for IgE PhIP-Seq were identical to the IgG PhIP-Seq methodology described above.

Sequencing reads were mapped to the AllerScan library, requiring perfect matches. We counted the number of times a clone was detected in each immunoprecipitate, creating a read count matrix. Next, we utilized the "edgeR" R software package, which compares the signal detected in each sample against a set of negative control "mock" immunoprecipitations that were performed without serum, using a negative binomial model, returning both a test statistic and fold-change value for each peptide in every sample, thus creating enrichment and fold-change matrices.

Significantly enriched peptides, called "hits", required counts, p values, and fold changes of at least 100, 0.001, and 5, respectively. Hits fold-change matrices report the fold-change value for "hits" and "1" for peptides that are not hits.

For the specific analysis of WOIT trial data ($N = 23$), we used a lenient threshold for response classification: requiring an increase of >1000 mg of wheat extract between baseline and after treatment to be considered a response.

This study compared reactivities from all placebo patients before and after placebo treatment and compared reactivities from all patients before and after 1 year of active treatment. Two patients did not complete the trial, two patients failed to respond by this metric, and the remaining 19 patients were considered responders.

All analyses were developed and implemented in R 3.6.1. All heatmaps were constructed using the pheatmap package and all network graph analyses utilized the igraph software package; all other plots were built using the tidyverse package suite. Heatmap clustering used the Euclidean distance measure and the Ward.D2 linkage measure. All statistical tests were two-sided Wilcoxon signed-rank tests unless otherwise specified.

**Reporting summary**. Further information on research design is available in the Nature Research Reporting Summary linked to this article.

## Data availability

The data that support the findings of this study are available in figshare with the identifier [https://doi.org/10.6084/m9.figshare.13315220.v1]. All analyses can be reproduced by running the publicly hosted dataset through the custom scripts available at https://github.com/drmonaco/AllerScan. Source data are provided with this paper.

## Code availability

All custom scripts used to analyze the data in this study can be found at https://github.com/drmonaco/AllerScan.

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

## Acknowledgements

This work was made possible by generous support from the National Institutes of Health, as a U24 grant (AI118633) from NIAID's Development of Sample Sparing Assays Program, and an R01 grant (GM136724) from NIGMS. P.A.F. is supported by the Division of Intramural Research, NIAID, NIH. We are grateful to Stephen Elledge for providing the T7FNS2 phage vector, and to Uri Laserson for his invaluable assistance in the design of the AllerScan library.

## Author contributions

D.R.M., P.A.F., and H.B.L. conceived the project, and wrote the manuscript. D.R.M., B.M.S., and T.R.N performed the PhIP-Seq testing of the serum samples. D.R.M. developed the software for analysis of the PhIP-Seq data. H.A.S., A.N.-W, and R.A.W. provided the wheat oral immunotherapy samples and conducted preliminary serum testing. P.A.F. provided and characterized all other samples. A.C.K. and R.G.H. provided valuable insight on the manuscript. All authors provided critical feedback and helped shape the research, analysis, and manuscript.

## Competing interests

D.R.M. and H.B.L. are listed as inventors on a patent describing the AllerScan technology. R.A.W. receives research support from Aimmune, Astellas, DBV, Genentech, HAL-Allergy, Regeneron, and Sanofi. H.A.S. receives consulting fees from N-Fold Therapeutics, DBV Technologies, and Siolta Therapeutics. Other authors declare no competing interests.
