## [Peer Review File · Nature Communications]

REVIEWERS' COMMENTS

Reviewer #1 (Remarks to the Author):

My prior review of this manuscript has been addressed fully and satisfactorily.

I have re-read the manuscript, with particular attention the authors' rebuttal to the comments of the two others reviewers, and conclude that here too there has been a full and satisfactory reply.

The inclusion of the new Supplemental Figure 2 is an excellent addition, and adds even more to the wealth of informative data included in the manuscript.

Reviewer #2 (Remarks to the Author):

The diagnosis of food allergy by testing for specific IgE is fraught with poor specificity. Many subjects with IgE to a food can ingest it without having allergic reactions. Although a number of factors might contribute to the weak correlation between the presence of allergen specific IgE and allergic reactions on ingestion, the one that is considered in this study is that the presence of IgE antibodies directed at specific epitopes might favor reactions while, conversely, IgG's directed at certain epitopes might inhibit reactions. For the most part, in vitro testing for allergen specific antibodies uses immobilized crude allergen extracts containing many component proteins. For some allergens, tests have been developed at the individual protein ("component") level and some of these have been helpful. Peptide based approaches have been piloted but are technically limited by the repertoire of peptides that can be analyzed.

Here Monaco and colleagues adapt a platform of phage immunoprecipitation adapted from one previously reported by the senior author and colleagues as "Virscan" and now named "AllerScan" to identify IgE- and IgG-binding wheat allergen epitopes at 56-mer peptide resolution. This involves construction of a T7 bacteriophage library, using synthetic oligonucleotides based on the Allergome database, incubation of phage with patient sera and immunoprecipitation with anti-IgE or protein-G magnetic beads, followed by sequencing. They apply AllerScan to sera from patients with wheat allergy (IgE+ and reactions) vs sensitization (IgE+ but no clinical reaction) and observe that IgE from allergic subjects binds to a broad range of epitopes, some corresponding to previously reported peptide epitopes for wheat. Less reactivity was seen with wheat specific IgE from sensitized subjects. Allergic subjects exhibited reactivity with more independent epitopes. IgG epitopes largely overlapped with IgE. A peptide from alpha purothionin was preferentially bound by IgG from sensitized and nonallergic subjects vs. allergic subjects. AllerScan is also applied to undergoing wheat oral immunotherapy (OIT) and, as expected induces a several log increase in IgG binding to many wheat epitopes including alpha purothionin.

The adaptation of phage immunoprecipitation to allergen epitope identification using IgE antibody IP is innovative and will have impact on the field. The authors have convincingly demonstrated the ability to identify and track reactivity of both IgE and IgG antibodies to a large range of peptides in an unbiased manner. They report overlap with reported target peptides and the IgG responses during OIT are as predicted. The specific association of IgE: alpha purothionin reactivity and clinical allergy is of

great interest and is consistent with published reports by Valenta and colleagues that this Tri a 37 correlates with severe wheat allergy.

While the ability of AllerScan to display long peptides is an advantage it has the corresponding disadvantage of relatively low resolution. Mutagenesis would greatly sharpen the focus but is not employed in the current study. Of course, some of the most important wheat allergen epitopes are likely conformational and therefore, unfortunately, invisible in a peptide display approach.

Overall this is an important and well-done study. It represents a technical advance in allergen epitope analysis and shows that even minute amounts of IgE, as are normally present in plasma, can be used in an immunoprecipitation strategy. With respect to novel findings re wheat allergy, the advances are modest.

Reviewer #3 (Remarks to the Author):

I have reviewed this manuscript earlier and recommended publication of the paper after revision. The authors have addressed now several of my comments. In principle the revised paper is now improved although it would have been nice if the authors would have elaborated some points in more depth such as quoting the prevalence of food allergy and the discussion of their findings in the context of isotype switch towards IgE.